# Staphylococcus Aureus Bacteriuria as a Predictor of In-Hospital Mortality in Patients with Staphylococcus Aureus Bacteremia. Results of a Retrospective Cohort Study

**DOI:** 10.3390/jcm9020508

**Published:** 2020-02-13

**Authors:** Tobias Siegfried Kramer, Beate Schlosser, Désirée Gruhl, Michael Behnke, Frank Schwab, Petra Gastmeier, Rasmus Leistner

**Affiliations:** 1Institute of Hygiene and Environmental Medicine, Charité Universitätsmedizin Berlin, Hindenburgdamm 27, 12203 Berlin, Germany; beate.schlosser@charite.de (B.S.); desiree.gruhl@charite.de (D.G.); michael.behnke@charite.de (M.B.); frank.swab@charite.de (F.S.); petra.gastmeier@charite.de (P.G.); rasmus.leistner@charite.de (R.L.); 2National Reference Center for the Surveillance of Nosocomial Infections, 12203 Berlin, Germany

**Keywords:** Staphylococcus aureus, bacteremia, bloodstream infection, bacteriuria

## Abstract

Staphylococcus aureus bloodstream infection (SA-BSI) is an infection with increasing morbidity and mortality. Concomitant Staphylococcus aureus bacteriuria (SABU) frequently occurs in patients with SA-BSI. It is considered as either a sign of exacerbation of SA-BSI or a primary source in terms of urosepsis. The clinical implications are still under investigation. In this study, we investigated the role of SABU in patients with SA-BSI and its effect on the patients’ mortality. We performed a retrospective cohort study that included all patients in our university hospital (Charité Universitätsmedizin Berlin) between 1 January 2014 and 31 March 2017. We included all patients with positive blood cultures for Staphylococcus aureus who had a urine culture 48 h before or after the first positive blood culture. We identified cases while using the microbiology database and collected additional demographic and clinical parameters, retrospectively, from patient files and charts. We conducted univariate analyses and multivariable Cox regression analysis to evaluate the risk factors for in-hospital mortality. 202 patients met the eligibility criteria. Overall, 55 patients (27.5%) died during their hospital stay. Cox regression showed SABU (OR 2.3), Pitt Bacteremia Score (OR 1.2), as well as moderate to severe liver disease (OR 2.1) to be independent risk factors for in-hospital mortality. Our data indicates that SABU in patients with concurrent SA-BSI is a prognostic marker for in-hospital death. Further studies are needed for evaluating implications for therapeutic optimization.

## 1. Introduction

Staphylococcus aureus bacteremia (SA-BSI) is a severe and frequent bacterial infection [1]. The mortality of patients remains elevated despite effective diagnostic and therapeutic options, even in highly developed healthcare settings [2]. Several official guideline and recommendations on management of SA-BSI are available [2,3,4]. The German Society of Infectious Diseases has published recommendations on the diagnosis and treatment of SA-BSI to improve the care and outcomes of patients [3]. Since the late 1970′s, several reports have been published regarding the clinical significance of Staphylococcus aureus bacteriuria (SABU) in patients with SA-BSI [5]. The role of SABU described in the literature covers a wide range of potential implications for patients from asymptomatic colonization to primary catheter associated urinary tract infection to hematogenous seeding in patients with SA-BSI [6,7]. While certain toxins were shown to play a role in severity of SA-BSI [8], the invasion of host cells and virulence of S. aureus is largely determined by fibronectin binding and it could explain intrarenal translocation into the urinary tract [9]. A recent in-vivo study suggested that urinary tract colonization and infection with MRSA is based on the release of fibrinogen after catheter associated tissue damage in mice and humans [10]. In relation to that, SABU was described as a prognostic factor in the identification of complicated SA-BSI [11], as well as being associated with increased mortality in patients with SA-BSI [12,13]. A recent report found that only few patients with a SABU had a concomitant SA-BSI (6.9%) [7]. Moreover, a national study from Iceland showed that, after excluding primary SABU, 10.5% of patients with SA-BSI still developed secondary SABU [14]. Nevertheless, the clinical interpretation of this situation remains controversial [15]. Therefore, our objective was to describe the potential role of SABU in patients with SA-BSI and evaluate its influence on in-hospital mortality.

## 2. Experimental Section

### 2.1. Setting, Study Design and Data Collection

The study was conducted at three different tertiary care hospitals of Charité Universitätsmedizin Berlin, with 3011 beds in total. Prior to the study, we obtained an official ethics-vote from the Charité Universitätsmedizin ethics committee (internal processing key EA2/130/19). The investigation was performed as a retrospective cohort study that aimed to include all cases of SA-BSI in patients 18 years of age or older, between 1 January 2014 and 31 March 2017. Cases were identified in the Charité microbiology database as patients with blood cultures positive for Staphylococcus aureus who had had urine that was sampled within 48 h prior or after first S. aureus-positive blood culture. For all of the patients enrolled in this study, the following demographic and clinical characteristics were collected: age, sex, in-hospital death, length of hospital stay (LOS), day of BSI onset, and stay on an intensive care unit (days). Length of stay in total and after BSI onset were defined as length of stay until death or discharge. We assessed the PITT Bacteremia Score on the day of BSI onset retrospectively from patient files in order to evaluate the severity of the bloodstream infection [16]. The Charlson comorbidity index (CCI) was obtained on the basis of the patients’ diagnosed comorbidities while using the method of Charlson et al. [17] and the adaptation for the ICD-10 by Thygesen et al. [18].

### 2.2. Definitions

Cases were defined as patients with SA-BSI during the study period. The onset of SA-BSI was defined as the date of the sampling of the first blood culture with growth of Staphylococcus aureus. SA-BSI was considered to be hospital-onset if it occurred after the second day of hospitalization (>48 h after admission). SABU was defined as a detection of Staphylococcus aureus in a urine sample. Urinary Catheter prior to onset was defined as an indwelling urinary catheter for any length of time within seven days prior to onset of SA-BSI. Effective antimicrobial treatment was defined as flucloxacillin or cefazolin i.v. treatment in MSSA and vancomycin, linezolid, daptomycin, or ceftaroline treatment in MRSA after lab results were available (72 h) [19,20]. Mortality was assessed based on discharge alive or in-hospital death.

### 2.3. Statistical Methods

Descriptive univariate analyses were performed for the total cohort, being stratified by in-hospital death and by SABU. The median and the interquartile range (IQR) were calculated for continuous parameters; the number and percentage were calculated for binary parameters. Univariate differences were tested while using the Wilcoxon rank-sum test for continuous variables and the Chi-square test for binary variables. All tests of significance were two-tailed with a *p*-value of <0.05 considered to be significant. Univariate survival analysis was performed as a Kaplan–Meier curve stratified by SABU.

We performed a Cox-proportional hazard regression adjusting for LOS after BSI in order to analyze potential factors on in-hospital mortality in a multivariable analysis. The variables included in the analysis were the ones with a *p*-value < 0.100 in the univariate analysis and potential influence on the endpoint: SABU, Pitt-Score, ICU admission after onset of SA-BSI, urinary catheter prior to sampling, peptic ulcer, diabetes with and without complication, renal disease, and moderate to severe liver disease. As the duration of antimicrobial therapy depended on the LOS after BSI onset, it was not included. We calculated adjusted hazard ratios (HR) with a 95% confidence interval for in-hospital death after onset of SA-BSI while using a stepwise forward approach. For the final model, variables with *p*-values of ≤0.05 were included and variables with *p* > 0.05 were excluded while using a stepwise forward approach. All of the analyses were performed while using SPSS (IBM SPSS statistics, Somer, NY, USA) and SAS (SAS Institute, Cary, NC, USA).

### 2.4. Microbiological Methods

Urine sampling was performed with native mid-stream or catheter urine in a standard 10mL sterile Urin-Monovette^®^ (Sarstedt, Nümbrecht, Germany). The samples were cultured for up to forty-eight hours. If a blood stream infection was suspected, the blood cultures were drawn and incubated for up to seven days while using standard blood culture bottles (BACTEC^®^, Becton Dickinson, Heidelberg, Germany). Gram staining and subculturing were performed if growth was detected. MALDI TOF MS^®^ and Vitek 2^®^ automated system (Biomerieux, Marcy l’etoile, France) were used for identification and susceptibility testing of bacterial strains. They were interpreted while using EUCAST definitions.

## 3. Results

In the study period, 1139 patients were diagnosed with S. aureus bacteremia. In addition, 206 patients also had at least one urine culture examination within 48 h before or after SA-BSI. Of these patients, we had complete datasets of 202 patients (Figure 1). The remaining patients had to be removed due to incomplete data.

In this cohort (Table 1), the overall in-hospital mortality after onset of SA-BSI was 27.23% (*n* = 55). Univariate analysis showed that patients that died during their treatment in hospital had a median PITT bacteremia score of two (IQR, 1–8). In comparison, patients that were discharged from hospital had a significantly lower score of one (IQR, 0–2). Patients that died during their hospital stay were more likely to have been admitted to an ICU after the onset of SA-BSI when compared to surviving patients. They also had shorter LOS overall and stayed for shorter periods after the onset of SA-BSI (Table 2). Their antimicrobial treatment was shorter and they were less likely to receive antimicrobial treatment for >14 days (Table 2 and Appendix A). However, they were more likely to have a urinary catheter prior to onset of SA-BSI and were more likely to have urine cultures that were positive for S. aureus (Table 2).

We stratified the cohort into groups with an *S. aureus*-negative (SA-negative) or an *S. aureus*-positive (SA-positive) urine culture (Table 3 and Appendix A). Univariate analysis in this comparison revealed that patients with SA-positive urine had a higher mean CCI and higher in-hospital mortality than patients with SA-negative urine. The mean onset of SA-BSI was earlier (day 1; IQR 0–7) than in patients with SA-negative urine (day 4; IQR 1–11). Patients with SA-positive urine had a shorter LOS overall and shorter LOS after the onset of SA-BSI.

Univariate survival curve (Kaplan–Meier) showed a higher risk of death in patients with SA-BSI and concurrent SABU than those with SA-BSI only (Figure 2).

In a Cox regression analysis, we took the length of hospital stay into consideration. In this analysis, a Pitt Bacteremia Score of >1, SA-positive urine culture, as well as moderate to severe liver disease, were independent risk factors (Table 4 and Figure 3).

Displayed is the multivariable hazard regression function of patients with S. aureus bloodstream infection that is stratified by positive urine culture and adjusted for length of stay after onset of SA-BSI, Pitt bacteremia score, SABU, S. aureus bacteriuria. SA-BSI, S. aureus bloodstream infection. LOS, length of stay.

## 4. Discussion

We identified SABU as an independent and clinically relevant risk factor for in-hospital mortality in patients with SA-BSI. This finding is consistent with earlier reports [12,13]. However, the pathophysiological reason for SABU can vary and it is often not associated with SA-BSI [21].

SABU is considered to be a secondary seeding to SA-BSI [21]. Others question this theory and maintain that SABU mainly occurs because of the urinary catheter-associated contamination or primary infection [15]. A recent population-based study from Canada showed that only 7% of all patients with a SABU also develop a SA-BSI [7]. While there were urine samples that were derived from catheterized patients in our cohort, the rate of catheterization prior to BSI onset was similar for patients with and without SABU. Hence, contamination as an underlying reason seems to be less likely. This is supported by the results of earlier studies. Lafon et al. identified community-acquired SABU to be associated with left sided endocarditis in patients without risk for urinary tract colonization [22]. A recent Meta-analysis described SABU as an important predictor for SA-BSI and endocarditis in patients without urinary tract-infection or colonization [21]. In addition, nosocomial BSI were observed in higher frequencies among patients without SABU. This finding is potentially explained by the observation that nosocomial SA-BSI are often based on primary BSIs, which are frequently associated with vascular catheters [23]. Vascular catheter association are common in nosocomial SA-BSI and they have a lower mortality when compared to community acquired SA-BSI [24]. However, we conducted a multivariable analysis to evaluate the risk factors for SABU (Appendix A). The results showed that the development of SABU in our cohort was independent from the timing of BSI onset.

Our multivariable logistic regression identified moderate to severe liver disease to be an independent risk factor for in-hospital mortality after *S. aureus* bloodstream infection, which was previously known [22,25]. Moreover, Bassetti et al. described liver disease as a risk factor, especially in patients with community-acquired SA-BSI [26].

The Pitt Bacteremia Score showed an independent direct correlation with hospital mortality in patients with SA-BSI. This finding is in line with other studies [27,28]. Roth et al. found the Pitt Bacteremia Score to have a low positive predictive value, especially in short-term mortality for SA-BSI [29]. However, as this might be true in low mortality risk populations (their median Pitt Bacteremia Score was 0 and median CCI was 3), cohorts with an overall higher mortality risk yield a higher positive predictive value (PPV). The median Pitt Bacteremia Score in our cohort was 1, and the median CCI was 7, indicating a cohort with a high mortality risk.

In-hospital mortality of patients with SA-BSI was 27%. Similar in-hospital mortality rates have been described for patients with SA-BSI in comparable settings [30]. Hence, we assume that our study has certain representativeness. However, there are limitations to our study that have to be acknowledged: (i) This is a retrospective cohort study from a single university hospital center. Therefore, the selection of our cohort is influenced by the microbiological sampling strategies, especially in regards to urine samples. Our institutional guidelines give clear recommendations on sampling strategies to reduce contamination. It is likely that urine was sampled either when patients had symptoms suggestive of urinary tract infections or when patients were admitted with symptoms of a systemic infection that did not have a clear-cut focus. This certainly applies to patients that are admitted to a hospital because of a systemic infection. Additionally, we neither analyzed patients with SABU for whom we did not have blood culture samples, nor patients with SA-BSI with end stage renal disease that had anuria. (ii) We were unable to differentiate between the primary focus and secondary site in patients with positive urine, since we did not record specific clinical symptoms for UTI. However, this is not relevant to our objectives, since we focused on outcome parameters. (iii) *S. aureus* isolates were neither tested for relation nor virulence factors in order to account for differences. (iv) Only a small portion of patients received a treatment strategy the followed national and international guidelines. While it is necessary to validate our results after improvement in the management of patients with SA-BSI, the choices of substances and the duration of antimicrobial treatment were independent of urine sample results.

## 5. Conclusions

Our study demonstrates that SABU is a relevant prognostic marker for in-hospital mortality in patients with SA-BSI. This suggests a potential benefit from intensified treatment strategies, as well as from efforts to identify the primary source of SA-BSI. Further clinical studies are needed to confirm this finding.

## Figures and Tables

**Figure 1 jcm-09-00508-f001:**
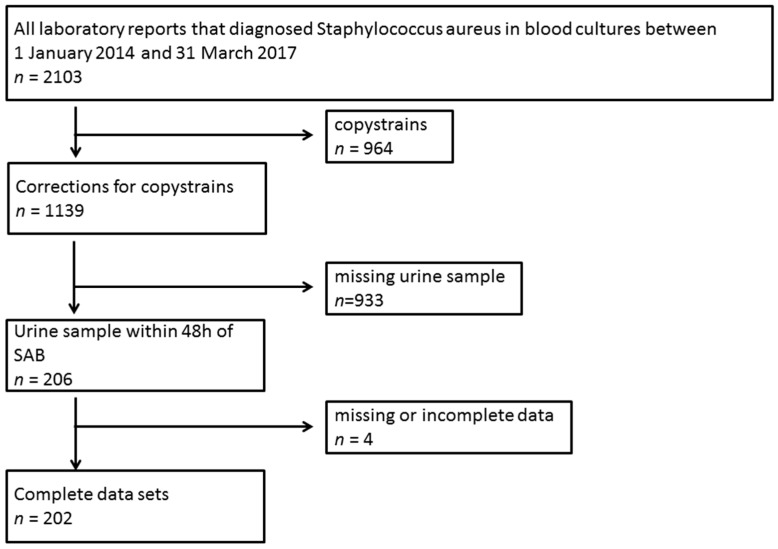
Flowchart depicting patient recruitment based on blood culture isolates and urine testing.

**Figure 2 jcm-09-00508-f002:**
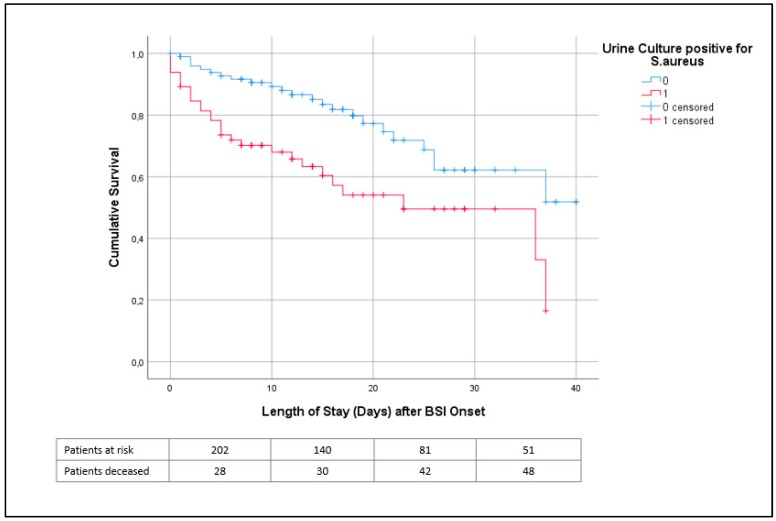
Univariate survival curve (Kaplan Meier curve) of patients with *S. aureus* BSI Displayed is the survival curve of patients with *S. aureus* bloodstream infection stratified by positive urine culture. The curve is restricted to a maximum observation time (length of stay) of 40 days. Censored were patients that left the hospital alive.

**Figure 3 jcm-09-00508-f003:**
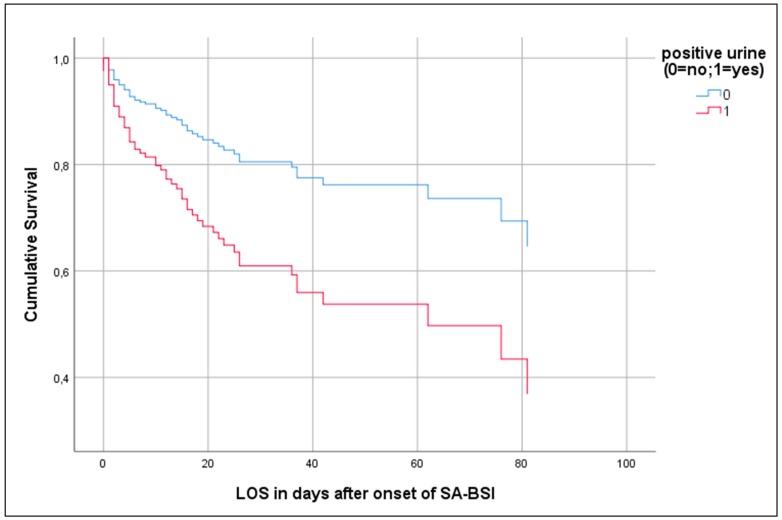
Cox proportional hazards regression of patients with SA-BSI.

**Table 1 jcm-09-00508-t001:** Demographic data on the study cohort.

	Total *N* = 202
	Median (IQR)/% (*n*)
Age in years	71 (59–80)
Male gender	60% (121)
in-hospital death	27% (55)
Onset of SA-BSI after admission in days	2 (1–10)
LOS in days	22 (12–42)
ICU admission after onset of SA-BSI	61% (124)
SABU	39% (78)
SA-BSI with MRSA	25% (50)
Pitt-Score	1 (0–4)
Charlson comorbidity index	7 (5–10)
Effective antimicrobial treatment	42% (86)
Spondylodiszitis	2% (4)
Endocarditis	8% (17)
Cardiac device	6% (13)
Artificial heart valve	6% (13)
Dialysis prior to sampling	15% (30)
Port catheter prior to sampling	9% (19)
CVC prior to sampling	31% (63)
Urinary catheter prior to sampling	49% (98)

Continuous variables are presented as median and interquartile range (IQR). Categorical variables are presented as number (%). BSI: bloodstream infection, ICU: intensive care unit, LOS: length of stay, and SABU: *S. aureus bacteriuria*.

**Table 2 jcm-09-00508-t002:** Univariate analysis of the study cohort stratified by in-hospital death.

	Total *N* = 202	Discharged Alive *N* = 147	In-Hospital Death *N* = 55	*p*-Value
	Median (IQR)/% (*n*)	Median (IQR)/% (*n*)	Median (IQR)/% (*n*)	
LOS in days	22 (12–42)	28 (15–47)	13 (5–28)	<0.001
LOS after SA-BSI onset in days	16 (8–30)	18 (11–37)	8 (2–19)	<0.001
Pitt-Score	1 (0–4)	1 (0–2)	2 (1–8)	<0.001
SABU	39% (78)	33% (49)	53% (29)	0.012
Peptic ulcer	4% (9)	6% (9)	−	0.060
Diabetes without complication	28% (56)	32% (47)	16% (9)	0.027
Diabetes with complication	9% (19)	12% (17)	4% (2)	0.086
Renal disease	62% (126)	59% (86)	73% (40)	0.063
Moderate to severe liver disease	13% (27)	10% (14)	24% (13)	0.009
ICU admission after onset of SA-BSI	61% (124)	55% (81)	78% (43)	0.003
Urinary catheter prior to sampling	49% (98)	44% (65)	60% (33)	0.058

Only the relevant results with a *p*-value ≤0.100 are shown, as they were further analyzed in the multivariable analysis. A table of the entire univariate analysis can be found in the supplement under S1. Continuous variables are presented as median and interquartile range (IQR). Categorical variables are presented as number (%). BSI: bloodstream infection, LOS: length of stay, CCI: Charlson comorbidity index. ABX: antimicrobial therapy. SABU: *S. aureus bacteriuria*.

**Table 3 jcm-09-00508-t003:** Univariate analysis of the study cohort stratified by urine positive or negative for Staphylococcus aureus.

	Negative Urine	Positive Urine/SABU	*p*-Value
	Median (IQR)/% (*n*)	Median (IQR)/% (*n*)	
Total	100% (124)	100% (78)	not applicable
Male gender	52% (65)	72% (56)	0.008
Onset of SA-BSI after admission in days	4 (1–11)	1 (0–7)	0.044
LOS in days	27 (15–45)	18 (7–39)	0.004
LOS after SA-BSI onset in days	18 (10–32)	14 (5–28)	0.018
Pitt-Score	11 (9–15)	10 (5–14)	0.062
Charlson comorbidity index	7 (5–9)	8 (6–10)	0.008
In-hospital mortality (0 = no;1 = yes)	21% (26)	37% (29)	0.012
Cardiac insufficiency	38% (47)	26% (20)	0.072
Connective tissue disease	4% (5)	−	0.073
Malign tumor	13% (16)	31% (24)	0.002
Metastatic solid tumor	13% (16)	24% (19)	0.036
Hemiplegia	7% (8)	15% (12)	0.038
Lymphoma	10% (12)	3% (2)	0.053

Shown are only the results with a *p*-value ≤ 0.100. A table of the entire univariate analysis can be found in the supplement under S2. Continuous variables presented as median and interquartile range (IQR). Categorical variables are presented as number (%). BSI: bloodstream infection. LOS: length of stay, CCI: Charlson comorbidity index. ABX: antimicrobial therapy.

**Table 4 jcm-09-00508-t004:** The results of cox regression for in-hospital death after *S. aureus* bloodstream infection.

	*p*-Value	Odds Ratio	CI95 Low	CI95 High
SABU	0.003	2.281	1.224	3.899
Pitt bacteraemia score	<0.001	1.195	1.107	1.289
Moderate to severe liver disease	0.023	2.062	1.104	3.853

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
