# Peer review of "Staphylococcus Aureus Bacteriuria as a Predictor of In-Hospital Mortality in Patients with Staphylococcus Aureus Bacteremia. Results of a Retrospective Cohort Study"

_jcm, 2020, doi:10.3390/jcm9020508_

Round 1
Reviewer 1 Report
Acceptable cohort despite large proportion of SA-BSI patients that did not have an urine culture collected within 48 hours pre- or post SA-BSI onset with possible selection bias preventing generalizability of the study result.
Definitions of effective antibiotic in background questionable. Antibiotics marked as “S” by EUCAST criteria effective against S aureus although not recommended in SA-BSI guidelines, for example the MRSA drugs are also effective against MSSA, other cephalosporines (except ceftazidim, ceftibuten) effective against MSSA etc.
Basic demographic data like age and male/female proportion missing in table 1, necessary to describe type of cohort, comparability to other study cohorts. Demographic and outcome data mixed. LOS in patients with in-hospital death shorter which is expected due to survival bias. Longer treatment in survival group also influenced by survival bias. Treatment duration significantly shorter than German recommendations in surviving patients. Follow up oral antibiotics or other missing data?
Table 2: number of patients in each column missing (N=126, N=78). Significantly more nosocomial SA-BSI patients among non-SABU patients. Nosocomial onset of SA-BSI good prognostic factor, correlates to early treatment? Comment and data needed.
Statistical methods and results overused and interpreted. Not relevant to present both a multivariable logistic regression and a Cox regression for in-hospital mortality
Specific comments:
Line 70. Define by hours (48h?), unclear meaning of “after the second day of hospitalization”
Line 84 and 86 (and 90) : What p-value in the univariable analyses was used for inclusion in the multivariable model, ≤ 0,05 or ≤0.100? Clarify how the logistic regression models were constructed.
Line 86: Use the word univarbiale or univariate consequently?
Discussion:
Can be shortened and focused
Not possible to follow reference to quoted publications, for example line 168-171.
Study limitations relevantly presented.
Author Response
Comments and Suggestions for Authors
Acceptable cohort despite large proportion of SA-BSI patients that did not have an urine culture collected within 48 hours pre- or post SA-BSI onset with possible selection bias preventing generalizability of the study result.
Definitions of effective antibiotic in background questionable. Antibiotics marked as “S” by EUCAST criteria effective against S aureus although not recommended in SA-BSI guidelines, for example, the MRSA drugs are also effective against MSSA, other cephalosporines (except ceftazidim, ceftibuten) effective against MSSA etc.
Authors reply:
We fully agree with your statement in general, but recommended substances for SA-BSI are limited and associated with an improved outcome in patient. Therefore we decided to use this stricter definition. However; empiric treatment usually included a broad-spectrum approach, we tried to account for proper treatment after results for pathogen identification and susceptibility were available.
Basic demographic data like age and male/female proportion missing in table 1, necessary to describe type of cohort, comparability to other study cohorts. Demographic and outcome data mixed.
Authors reply:
Thank you for this remark. We added an additional table with demographics of the cohort (Table 1). Information on gender was added as well to the tables and the analysis.
LOS in patients with in-hospital death shorter, which is expected due to survival bias. Longer treatment in survival group also influenced by survival bias.
Authors reply:
We double checked and performed a linear regression in order to find the factors influencing the lengths of stay (see table below). Indeed mortality shortened LOS significantly. Therefore it is also a confounder for length of ABX treatment. Therefore we removed length of ABX treatment from the analysis and adjusted only for LOS in a cox regression.
Treatment duration significantly shorter than German recommendations in surviving patients. Follow up oral antibiotics or other missing data?
Authors reply:
This is an important remark. The overall short treatment duration is a reflection of a lack of consultations prior to establishing the local antimicrobial stewardship team. Even though we did not account for treatment options post discharge, its influence might be limited for our cohort since no Outpatient antimicrobial therapy was available at our center during the study period and oral treatment has not yet been sufficiently studied in SA-BSI. Nonetheless we removed length of ABX from our analysis due to the limitations stated in the previous comment.
Table 2: number of patients in each column missing (N=126, N=78).
Authors reply:
The absolute numbers have been added to the table.
Significantly more nosocomial SA-BSI patients among non-SABU patients.
Nosocomial onset of SA-BSI good prognostic factor, correlates to early treatment? Comment and data needed.
Authors reply:
This is a very relevant observation, which we added to our analysis (supplement table 3) and the discussion (Lines 174-180). Nosocomial SA-BSI are often based on primary BSIs, which are frequently associated with vascular catheters (Kaech C). Vascular catheter association are common in nosocomial SA-BSI and have a lower mortality when compared to community acquired SA-BSI (Laupland KB). Multivariable analysis showed that development of SABU is independent from onset after hospital admission (supplement Table 3).
Added Literature reference.
Kaech C, Elzi L, Sendi P, et al. Course and outcome of Staphylococcus aureus bacteraemia: a retrospective analysis of 308 episodes in a Swiss tertiary-care centre. Clinical microbiology and infection : the official publication of the European Society of Clinical Microbiology and Infectious Diseases. 2006;12(4):345-352.
Laupland KB, Church DL, Mucenski M, Sutherland LR, Davies HD. Population-based study of the epidemiology of and the risk factors for invasive Staphylococcus aureus infections. The Journal of infectious diseases. 2003;187(9):1452-1459.
Statistical methods and results overused and interpreted. Not relevant to present both a multivariable logistic regression and a Cox regression for in-hospital mortality
Authors reply:
We fully agree, used only the Cox regression analysis to adjust for LOS and removed the multivariable log regression in order to reduce over usage and over interpretation as suggested.
Specific comments:
Line 70. Define by hours (48h?), unclear meaning of “after the second day of hospitalization”
Authors reply:
We added (>48h after admission). However, it is difficult to define exact timing in hours, since we do not account for the exact time of admission.
Line 84 and 86 (and 90): What p-value in the univariable analyses was used for inclusion in the multivariable model, ≤ 0,05 or ≤0.100? Clarify how the logistic regression models were constructed.
Authors reply:
We changed the statistics session following your remarks: In order to analyze potential factors on in-hospital mortality in a multivariable analysis we performed a Cox-proportional hazard regression adjusting for LOS after BSI. Variables included in the analysis were the ones with a p-value <0,100 in the univariate analysis and potential influence on the endpoint: SABU, Pitt-Score, ICU admission after onset of SA-BSI, urinary catheter prior to sampling, peptic ulcer, diabetes with and without complication, renal disease, moderate to severe liver disease. As duration of antimicrobial therapy depended on the LOS after BSI onset, it was not included. We calculated adjusted hazard ratios (HR) with 95% confidence interval for in-hospital death after onset of SA-BSI using a stepwise forward approach. For the final model, variables with p-values of ≤0.05 were included and variables with p>0.05 were excluded using a stepwise forward approach. All analyses were performed using SPSS (IBM SPSS statistics, Somer, NY, USA) and SAS (SAS Institute, Cary, NC, USA).
Line 86: Use the word univarbiale or univariate consequently?
Authors reply:
Thank you for the comment. We changed univariable to univariate throughout the entire manuscript.
Discussion:
Can be shortened and focused
Authors reply:
Thank you for your suggestion. We thoroughly reworked the discussion and deleted segments that were no longer relevant to our results, therefore shortened, and focused as suggested.
Not possible to follow reference to quoted publications, for example line 168-171.
Authors reply:
We rephrased this section and incorporated alternative literature into our list of references. We hope that this improved our line of thought.
Study limitations relevantly presented.
Reviewer 2 Report
Staphylococcus aureus bacteremia (SAB) is a common and important infection all over the world. In this manuscript, the authors provided a retrospective cohort study on the role of Staphylococcus aureus bacteriuria (SABU) as a predictor of mortality. The study included a large number of patients with Staphylococcus aureus bloodstream infection (SA-BSI) and has considerable clinical importance. However, one concern comes from the age of patients. As shown in Table 1, the median age in the discharged alive group is 4 years younger than the one in the in-hospital death group, although there is no significant difference. It might affect the result/conclusion as it is known that age is the most consistent and strongest predictor of infection-related mortality. In this case, it would be better to show the survival cure stratified by both positive urine culture and grouped age.
Author Response
Staphylococcus aureus bacteremia (SAB) is a common and important infection all over the world. In this manuscript, the authors provided a retrospective cohort study on the role of Staphylococcus aureus bacteriuria (SABU) as a predictor of mortality. The study included a large number of patients with Staphylococcus aureus bloodstream infection (SA-BSI) and has considerable clinical importance.
However, one concern comes from the age of patients. As shown in Table 1, the median age in the discharged alive group is 4 years younger than the one in the in-hospital death group, although there is no significant difference. It might affect the result/conclusion as it is known that age is the most consistent and strongest predictor of infection-related mortality. In this case, it would be better to show the survival cure stratified by both positive urine culture and grouped age.
Authors reply:
We performed SABU/non-SABU stratified Kaplan Maier analyses for 4 age strata. As you can see in every strata (Group1: 20-40 years; group2: 40-60, group3: 60-80, group4: 80-100years) SABU is associated with reduced chances of survival in the groups 2-4. In Group 1 no deaths were recorded. We chose not to include this analysis in the paper. However, we could add these in the supplementary section of our manuscript if you think this would improve the overall quality.
Round 2
Reviewer 1 Report
My concern has been addressed properly.
Author Response
Dear reviewer,
thank you for your valuable input. The manuscript was rechecked by a native speaker.
Reviewer 2 Report
This is a well written paper and the topic is interesting.
I think a limitation that should be added is that the method of sampling urine from indwelling urinary catheters must be considered; significant contamination of the specimen is possible depending on technique. It would also be interesting to know why urine was analyzed in patients with ESRD-these samples are of limited value.
Author Response
Dear reviewer,
thank you for your valuable input. The manuscript was rechecked by a native speaker.
I think a limitation that should be added is that the method of sampling urine from indwelling urinary catheters must be considered; significant contamination of the specimen is possible depending on technique.
We fully agree with your point that indwelling urinary tract catheters of cause false positive microbiological results due to contamination. They are also the cause of excess CAUTI, which are an underlying reason for bacteremia. All of our patients had underlying S. aureus bacteremia, contamination with S. aureus is rarely observed when sampling in catheterized patients according to our protocol (alcoholic disinfection of membrane, sterile cannula and syringe). Furthermore frequencies of catheterization were similar in different groups.
We have added the following parts further addressing this topic.
Line 201: Our institutional guidelines give clear recommendations on sampling strategies to reduce contamination.
Line172: While there were urine samples derived from catheterized patients in our cohort, the rate of catheterization prior to BSI onset was similar for patients with and without SABU. Hence, contamination as an underlying reason seems less likely.
It would also be interesting to know why urine was analyzed in patients with ESRD-these samples are of limited value.
This is a very important point we have tried to solve ever since planning this study. We do agree that patients ESRD are probably underrepresented in this study and the influence of severe renal diseases is probably underestimated. However the objective of this study focused on the importance of SABU in patients. While many patients with ESRD have anuria, these were not included due to a lack of samples. Patients with ESRD and urological complications were included. In our opinion this expands the generality of our findings in clinical care.
We have added the following to our limitations section.
Line 205: Also, we neither analyzed patients with SABU for whom we did not have blood culture samples, nor patients with SA-BSI with end stage renal disease that had anuria.